# The Improved Brain-Targeted Drug Delivery of Edaravone Temperature-Sensitive Gels by Ultrasound for γ-ray Radiation-Induced Brain Injury

**DOI:** 10.3390/pharmaceutics14112281

**Published:** 2022-10-25

**Authors:** Qian Li, Yizhi Zhang, Jinglu Hu, Bochuan Yuan, Pengcheng Zhang, Yaxin Wang, Xu Jin, Lina Du, Yiguang Jin

**Affiliations:** 1School of Pharmacy, Shandong University of Traditional Chinese Medicine, Jinan 250355, China; 2Department of Pharmaceutical Sciences, Beijing Institute of Radiation Medicine, Beijing 100850, China; 3School of Pharmacy, Henan University, Kaifeng 475004, China; 4Beijing Tiantan Hospital, Capital Medical University, Beijing 100070, China

**Keywords:** radiation-induced brain injury, temperature-sensitive gels, edaravone, ultrasound, behavior tests

## Abstract

Radiation-induced brain injury (RBI) is a common neurological disease caused by ionizing radiation (IR). Edaravone (EDA) is a free radical scavenger, has the potential to treat RBI. EDA loaded temperature-sensitive gels (TSGs) were prepared for subcutaneous injection to improve inconvenient administration of intravenous infusion. RBI mice model was established by irradiation of ^60^Co γ-ray on head. EDA TSGs could improve spontaneous behavior, learning and memory and anxiety of RBI mice by behavior tests, including the open field test, the novel object recognition test, the elevated plus maze test and the fear conditioning test. The therapeutic effects were enhanced with the assistance of ultrasound. Alleviative pathological changes, decreased the expression of Molondialdehyde (MDA) and Interleukin-6 (IL-6) in the hippocampus of brain, indicated reduced oxidative stress and inflammatory response with the treatment of EDA TSGs and ultrasound. Moreover, ultrasound was superior to the use of EDA TSGs. Safe and effective EDA TSGs were prepared for RBI, and the feasibility of brain-targeted drug delivery enhanced by ultrasound was preliminarily demonstrated in this study.

## 1. Introduction

Different populations have different sensitivity to ionizing radiation (IR) due to different genotypes [1]. Radio-sensitivity refers to the susceptibility of body, cells and tissues response to IR [2]. The sensitivity of normal tissues to IR is regulated by multiple genes involved in different cellular pathways, and the response of tissues to IR is strongly influenced by the DNA response of stem cells and progenitor cells [3,4]. Brain is one of the most sensitive areas to IR injury, which is prone to radiation-induced brain injury (RBI) [5]. However, it lacked effective treatment for RBI, so it is significant to find safe and effective novel preparations. This is of great significance to improve the survival rate and quality of life of patients treated with radiotherapy, and to protect the life safety of professionals affected by radiation [6]. 

Edaravone (EDA) is a free radical scavenger, which is clinically used to treat acute cerebral infarction [7] and acute ischemic stroke [8]. It was approved to treat amyotrophic lateral sclerosis (ALS) by the Food and Drug Administration (FDA) [9]. EDA can remove toxic oxygen free radicals that cause brain injury, inhibit oxidative damage of brain cells, vascular endothelial cells, and nerve cells, and improve the role of nerve dysfunction [10,11]. At present, EDA injection is the only existing dosage form for the treatment of acute cerebral ischemia and stroke. But the method of administration of EDA injection is intravenous infusion in clinical practice, which is not convenient to be used in emergent conditions.

In situ gels are preparations that can transform from liquid to non-chemical crosslinked semi-solid gel, due to the change in physiological environment (temperature, pH, ion) [12,13,14]. Temperature-sensitive gels (TSGs) are relatively mature and widely applied, which have certain adhesion and histocompatibility to extend residence time and improve the bioavailability [15,16]. TSGs can load different properties of drugs with simple preparation process and convenient use protocol [17]. 

Ultrasound, a potential physical technique, has been shown in studies that can enhance the delivery of chemotherapeutics [18] and formulations [19] across the BBB into the brain injury area. In our previous study, the effect of ultrasound opening BBB were confirmed and the appropriate parameters were optimized. Hence, the selected parameters were applied to assist EDA TSGs for RBI, to verify the feasibility of ultrasound enhanced drug brain-targeted delivery. 

Here, poloxamer 407 and poloxamer 188 were used as the gel substrates to prepare EDA temperature-sensitive gels (EDA TSGs) to increase the retention time in vivo and achieve sustained release, which is conducive to brain-targeted drug delivery. The pharmacodynamics of EDA TSGs was evaluated by using ^60^Co γ-ray to establish RBI model. Ultrasound is used to improve the brain-targeted delivery of EDA TSGs. The main evaluation indexes included behavioral analysis, H.E. pathological analysis, and the expression of oxidative stress index malondialdehyde (MDA) and inflammatory factor interleukin-6 (IL-6). EDA TSGs will provide a potentially effective treatment preparation for RBI, and ultrasound is expected to enhance drug brain-targeted delivery.

## 2. Materials and Methods

### 2.1. Materials

Poloxamer 407 and poloxamer 188 were purchased from Shanghai Yuanye Biotechnology Co., Ltd. (Shanghai, China). EDA was purchased from Shanghai Aladdin Biotechnology Co., Ltd. (Shanghai, China). EDA injection was purchased from Guorui Pharmaceutical Co., Ltd., Sinopharm Group (Shanghai, China). Medical ultrasonic coupling agent was purchased from Shenzhen Minhao Technology Co., Ltd. (Shenzhen, Guangdong, China). MDA determination kit was purchased from Nanjing Jiancheng Biological Engineering Research Institute (Nanjing, Jiangsu, China). Mouse IL-6 ELISA Kit was purchased from Shanghai Enzyme Linked Biotechnology Co., Ltd. (Shanghai, China).

### 2.2. Animals

Male C57BL/6J mice (20–22 g) were purchased from the SPF Biotechnology Co., Ltd. (Beijing, China). Principles of laboratory animal care were followed, and the animal operation procedures were performed in accordance with Animal Research: Reporting of In Vivo Experiments (ARRIVE) guidelines, as well as the Guide for the Care and Use of Laboratory Animals (Eighth Edition) (US National Institute of Health, Bethesda, MD, USA). The ethnical approval number was IACUC-DWZX-2020-532.

### 2.3. Ultrasound Processing

A dual-frequency ultrasound therapeutic device (UT1021, Dongdixin Technology Co., Ltd., Shenzhen, China) was used. It was composed of an ultrasound transducer, which is a single 35 mm-diameter discoid probe. Mice were anesthetized and prepared by removing scalp hair prior to sonication. The ultrasonic probe was tightly contacted with the head of each mouse by the medical ultrasonic couplant agent (M-250A6, Minhao Technology Co., Ltd., Shenzhen, China) between them. The ultrasonic parameters were set as follows: the frequency was 1 MHz, the intensity was 0.6 W/cm^2^, and the duration was 3 min.

### 2.4. Prescription Optimization of EDA TSGs

The gel matrix was prepared first by the cold solution method. Poloxamer 407 and poloxamer 188 (6:1, *w*/*w*) were added with 15 mL ultrapure water. After being stirred evenly, the mixture was swelled completely at 4 °C overnight. The drug powder of EDA and 30 mL ultrapure water were added to the blank TSGs with a final EDA concentration of 2 mg/mL. The EDA TSGs were swelled at 4 °C to obtain transparent gels with completely dissolved drug [20]. The concentration of poloxamer 407 and poloxamer 188 as the important factors, the gelation temperature and gelation time as the indexes to select the best prescription. 

### 2.5. Characterization of EDA TSGs

#### 2.5.1. Determination of Gelation Temperature and Gelation Time

The gelation temperature and gelation time of EDA TSGs as indexs of optimizing prescription was measured by the tube inversion method [21]. EDA TSGs of 2 mL were placed in the test tube and put in a thermostat water bath (HH-2, Zhiborui Instrument Manufacturing Co., Ltd., Changzhou, China). The temperature was slowly increased from 20 °C to 40 °C with the rate of 1 °C/min. The flowing condition of EDA TSGs was observed by inverting the test tube [22]. The state of gels different prescription was recorded at a typical temperature of 25 °C, 30 °C, 32 °C, 34 °C, 36 °C and 37 °C. The gelation temperature was defined as the temperature at which the liquid loses its fluidity, and the time of state transformation was defined as the gelation time. 

#### 2.5.2. Rheological Properties Investigation

The storage modulus (G′), the loss modulus (G″) and the compound viscosity (η) of EDA TSGs were determined with the oscillatory study of temperature and frequency respectively by advanced rotational rheometer (MCR301, Anton Paar Co., Ltd., Graz, Austria). The oscillation mode of temperature was as the followings: the strain amplitude was 0.05%, the shear frequency was 6.28 rad/s, the temperature range was 10–40 °C, the rate of increasing temperature was 1 °C/min. And the changes in the modulus and viscosity at body temperature (37 °C) within a frequency range of 0.1–100 rad/s were also measured [23].

#### 2.5.3. Syringe Ability of EDA TSGs

The syringe ability of EDA TSGs was investigated by injection. The pigments bright blue and lemon yellow were added into the EDA TSGs as the indicator colors, then EDA TSGs were injected into water and air (a glass plate exposed to air) of 37 °C and subcutaneous injection of mice [24]. The syringe ability and molding property of the EDA TSGs were observed. 

#### 2.5.4. The Micromorphology of EDA TSGs

The gelled EDA TSGs were put into liquid nitrogen to achieve brittle rupture. The freeze-dried sample was cut into the thin slices and pasted on the conductive adhesive. The surface was sprayed with gold, and the internal morphology of EDA TSGs was observed under the scanning electron microscope (SEM) (HITACHI S4800, Tokyo, Japan) [25]. 

### 2.6. Model Establishment of RBI and Administration Scheme

Mice were randomly divided into 6 groups with 10 mice each group, including the healthy group, the model group, the blank TSGs group, the EDA TSGs group, the ultrasoud assisted EDA TSGs group, and the EDA injection (the positive control) group.

The RBI modeling conditions were as the followings. After anesthesia, the mice were fixed on the operating plate with their head exposed, and the head was irradiated with ^60^Co γ-ray (15Gy, 60.41 cGy/min) at irradiation distance of 2.5 m [26]. The other parts of the mice were shielded with lead brick.

The healthy group: no operation; the model group: IR only; the blank TSGs: injected subcutaneously with 0.1 mL blank TSGs 2 h after IR; the EDA TSGs group: injected subcutaneously with 0.1 mL EDA TSGs 2 h after IR; the ultrasound assisted EDA TSGs group: given ultrasound (0.6 W/cm^2^, 3 min) on the mice head on Day 1 (2 h after IR) and Day 3, respectively and injected subcutaneously with 0.1 mL EDA TSGs after ultrasound application; the EDA injection group: intraperitoneally injected with 0.13 mL EDA injection (1.5 mg/mL); The treatment groups were administered once a day for consecutive 7 days. The open field test was performed on the Day 5, the novel-object recognition test was performed on Day 5, 6 and 7, the elevated plus maze test was performed on the 8th day, the fear conditioning test was performed on Day 9 and 10, and the biological tissue sample were collected on Day 11 (Figure 1).

### 2.7. Behavior Evaluation

#### 2.7.1. Spontaneous Behavior of Mice Evaluated by the Open Field Test

An open field instrument (ZS-KC, Zhongshi Dichuang Technology Development Co., Ltd., Beijing, China) was used to evaluate the autonomic responses, exploratory behavior and stress of mice in a new environment [27]. The area (25 cm × 25 cm) at the bottom center of the experimental box (50 cm × 50 cm × 35 cm) was defined as the central area, and the rest of the area was defined as the surrounding area. All mice were placed in the central area at the bottom (1/box only) to explore freely on Day 5 after administration. The attached software recorded the movement track automatically within 5 min. More importantly, 75% alcohol was sprayed to eliminate the influence of feces, urine and smell from the previous animal in all behavior tests.

The number of central entry, the central distance (mice distance in the central area) and the total distance (mice distance in all areas) of mice were used as statistical indicators. 

#### 2.7.2. Learning and Memory of Mice Evaluated by the Novel Object Recognition Test

The novel object recognition test evaluates learning and memory based on the principle that rodents have an innate tendency to explore new objects. It is carried out in the condition that experimental animal is completely free. The novel object recognition test uses the same equipment (ZS-KC, Zhongshi Dichuang Technology Development Co., Ltd., Beijing, China) as the open field test, but a different test mode. The process of test was as follows [28]. Period for adaptation: the mice were put into a field without objects to explore freely for 10 min on Day 5 and 6, in order to fully adapt to the experimental environment and reduce stress reaction; period for learning: the mice were placed in the field with two identical objects to explore freely for 10 min on Day 7, in order to become familiar with the objects; period for test: 1 h after period for learning, one of the old objects was replaced with a novel object in the field, and mice were free to explore for 5 min.

Recognition index = the exploring time of the novel object/(the exploring time of the novel object + the exploring time of the old object). The time spent exploring old and new objects was also included in the analysis.

#### 2.7.3. Anxiety of Mice Evaluated by the Elevated plus Maze Test

The elevated plus maze (ZS-DSG, Zhongshi Dichuang Technology Development Co., Ltd., Beijing, China) can be used to evaluate the anxiety of animal. The elevated plus maze of mice consisted of an open arm (60 cm × 6 cm) and a closed arm (60 cm × 6 cm × 15 cm) that crossed each other vertically, and the cross part was the central area (10 cm × 6 cm). The mice were placed at the central area to explore freely on Day 8, and the movement track within 5 min were recorded automatically by software [29].

The statistical indicators are the times of mice entering open arm and the ratio of open arm exploration (the exploring time of open arm/(the exploring time of open arm + the exploring time of close arm) 

#### 2.7.4. Reflex to Conditional Memory of Mice Assessed with the Fear Conditioning Test

The fear conditioning test evaluates learning and memory based on reflex to fear memories that result from repeated conditioned and unconditioned stimuli. The fear conditioning test is performed in a conditional fear box (Shanghai Jiliang Software Technology Co., Ltd., Shanghai, China). Firstly, the mice were placed in a conditioned fear box (1/box only) in turn and subjected to conditioned training. After 2 min of spontaneous activity, electric shock and sound were used to set up fear stimulation (bee buzzing for 20 s, electric shock of 0.5 mA for 2 s at the 18th s), which repeated for 5 times. After 24 h, the mice were placed in the same environment (without electric shock and sound stimulation) for spontaneous activity and the freezing time within 5 min was recorded [30,31]. The freezing behavior was defined as a static state in which breathing is the main activity without other activity.

The freezing time ratio (%) = the freezing time/total time × 100%. The total distance was also included in the analysis.

### 2.8. H.E. Staining of Mice Brain

The brains of 3 mice in each group were stripped, fixed in 4% PFA for a week before being embedded in paraffin and sectioned (4 mm). The dewaxed sections were stained with hematoxylin solution and eosin dye (H.E) before being dehydrated and sealed with neutral gum [32]. The final step was image acquisition and analysis with the software Case Viewer (2.3, Servicebio Technology Co., Ltd., Wuhan, China). 

### 2.9. Expression of MDA and IL-6 in the Hippocampus of Brain

The hippocampus of 3 mice in each group were collected and weighed accurately. Physiological saline was added according to the weight volume ratio (*w*/*v* = 1:10) and homogenized in a high-speed tissue grinder (KZ-11, Servicebio Technology Co., Ltd., Wuhan, China). The homogenate was centrifuged at 4 °C with 13,180× *g* for 10 min to obtain supernatant. Finally, the content of mMDA and IL-6 in the hippocampus were measured according to the standard protocol of the MDA assay kit and the mouse IL-6 ELISA kit. The OD values at 532 nm and 450 nm, respectively, were read by the microplate reader (ELX 800, Biotek Instrument Co., Ltd., Winooski, VT, USA). 

MDA (nmol/mg) = [(OD_Determination_ − OD_Control_) / (OD_Standard sample_ − OD_Blank_)] × concentration of standard sample (10 nmol/mL).

Calculation of IL-6 (pg/mL): the measured OD value was substituted into the standard curve drawn with the concentration and the OD value of the standard sample [33].

### 2.10. Data Analysis

Multiple groups comparison was analyzed with one-way ANOVA, and Least square method (LSD) in post hoc analysis was used to determine the significance. The datas are represented as the mean ± SD. *p* < 0.05 indicating statistical differences. All statistical analyses were performed using SPSS software (19.0, International Business Machines Corporation, Armonk, NJ, USA).

## 3. Results and Discussion

### 3.1. Optimal Prescription of EDA TSGs 

Poloxamer 407 aqueous solutions show thermoreversible properties, which are the basis of thermosensitive gel formation. The concentration of Poloxamer 407 affects the properties of the gel, such as gelation temperature, viscosity, and mechanical strength. Poloxamer 188, as excipients, promotes Poloxamer 407 action by optimising gelation temperature or increasing bioadhesive properties [34]. Gels with different concentrations of Poloxamer 407 at different temperatures were evaluated (Table 1).

To make the prepared EDA TSGs rapidly change into gels after subcutaneous injection and increase the retention time, the gelation temperature of gels is required to be close to the physiological temperature of 37 °C, and the gelation time < 2 min. Considering the above, the optimal prescription was selected is prescription 2: 20% poloxamer 407 and 3.3% poloxamer 188 (Figure 2).

### 3.2. Rheological Properties of EDA TSGs

#### 3.2.1. The Viscoelastic Modulus and the Compound Viscosity Varied with Temperature

The storage modulus (elastic modulus G′) represents the semi-solid state, and the loss modulus (viscosity modulus G″) represents the liquid state. When the temperature was 32.5 °C, the storage modulus G′ was equal to the loss modulus G″, which was the transition temperature of the EDA TSGs from the liquid to the semi-solid. The loss modulus G″ was higher than the storage modulus G′ in the range of 10 °C to 32.5 °C, indicating the flowing liquid state. Above 32.5 °C, the storage modulus G′ was greater than the loss modulus G″, which indicated that the EDA TSGs were semi-solid and showed gel elasticity. At near 32.5 °C, temperature-dependent changes of the viscoelastic properties showed great increase in both the elastic and viscosity moduli and reached a plateau after 32.5 °C (Figure 3A). The shorter the gelation time may face, the less elimination from the site of application, which was consistent with choice of a rapidly gelled formulation. 

#### 3.2.2. The Viscoelastic Modulus and Compound Viscosity Varied with Frequency

When the temperature was 37 °C and the shear frequency (ω) was in the range of 0.1–100 rad/s, the storage modulus G′ was always higher than the loss modulus G″, which indicated that the semisolid property of the EDA TSGs was stable (Figure 3B). Moreover, the viscosity of the gels decreased with the increased shear frequency, the shear thinning behavior represents good syringe ability of gels. 

### 3.3. Good Syringe Ability of EDA TSGs

EDA TSGs was injected into water of 37 °C to form the filamentous gel indicating good syringe ability of gels (Figure 4A). EDA TSGs was stable with fixed shape even if overturn in air of 37 °C (Figure 4B). A complete and stable EDA TSGs in mouse subcutaneous tissue, indicated that the gel was suitable for subcutaneous injection (Figure 4C).

### 3.4. Micromorphology of EDA TSGs

The method of brittle rupture in liquid nitrogen of polymer materials provided a real morphology under SEM. An interwoven honeycomb structure with the pore diameter of about 10 μm was showed inside the gel (Figure 5). The porous structure can be used as a repository for drugs loading and release. The pores size in the EDA TSGs was related to the cross-linking degree of the gel matrix. Dense crosslinks lead to small internal pores and slow substance exchange, which is conducive to reduce the leak of drugs. 

### 3.5. In Vivo Safety of Blank TSGs

Continuous injection of blank TSGs for 7 d. The H.E staining of heart, liver, spleen, lung, and kidney tissues of the blank TSGs group and the healthy group, which has no obvious difference (Figure 6). Heart clear nuclei in the middle and neat myocardial fibers, liver cell nucleus large and round, the spleen has obvious red pulp (containing red neutrophile granulocyte) and white pulp (containing many violet lymphocytes), alveolar structure is complete, kidney has a complete tubular and glomerular structure. No inflammatory cells, bleeding and other symptoms were observed, which proved that TSGs consisted of poloxamer 407 and poloxamer 188 had good histocompatibility and no toxicity in vivo.

### 3.6. Pharmacodynamics Evaluation

#### 3.6.1. The Open Field Test to Detect the Spontaneous Behavior

The open field test was used to evaluate the autonomic responses, exploratory behavior and stress reaction of the mice in a new environment. The spontaneous behavior of the model mice was less active than that of healthy mice, with most of the time spent in the surrounding area such as the edge and corner. For the EDA TSGs, ultrasound assisted EDA TSGs, and the EDA injection, all can increase the activity of mice, and no improvement was observed in the blank TSGs (Figure 7A). 

Compared with the healthy mice, the number of central entry (*p* < 0.001), the central distance (*p* < 0.001) and the total distance (*p* < 0.01) of mice in the model mice were significantly reduced, indicating the decreased autonomous exploration and spontaneous activities and increased anxiety. The EDA TSGs (*p* < 0.05) and the ultrasound assisted EDA TSGs (*p* < 0.05) significantly increased the number of central entry compared with the model mice, but still lower than that of the healthy mice (*p* < 0.05). The EDA injection significantly increased the number of times entering the center (*p* < 0.001), and there was no significant difference compared with that of the healthy mice (Figure 7B). The EDA TSGs, the ultrasound assisted EDA TSGs, and the EDA injection could significantly increase the central distance of mice (*p* < 0.001), and the effect of the ultrasound assisted EDA TSGs were better than that of the EDA TSGs (*p* < 0.05), but lower than that of the EDA injection (*p* < 0.05) (Figure 7C). Compared with the model mice, the total distance of the EDA TSGs (*p* < 0.01), the ultrasound assisted EDA TSGs (*p* < 0.001) and the EDA injection (*p* < 0.001) were significantly increased, but there was still a statistical difference compared with that of the normal (*p* < 0.01). The total distance of ultrasound assisted EDA TSGs was higher than that of the EDA TSGs (*p* < 0.05), but there was no significant difference compared with EDA injection (Figure 7D). The number of central entry, central distance and total distance of the blank TSGs were not improved, and the statistical differences were extremely significant compared with those of the healthy mice (*p* < 0.001).

All indicators from the open field test indicated that EDA TSGs, ultrasound assisted EDA TSGs, and EDA injection had significant effects on improving the autonomic responses, exploratory behavior, decreasing stress of the RBI mice. Among them, ultrasound assisted EDA TSGs were superior to the use of EDA TSGs alone.

#### 3.6.2. The Novel Object Recognition Test to Detect Learning and Memory 

The novel object recognition test was used to evaluate learning and memory. The healthy animals spent more time exploring the new object when they were presented with a familiar object and a new object. As shown in the track map of mice (Figure 8A), the healthy mice moved around the novel objects and explored the old objects less. The model mice had the nearly same time of exploration for the old and the novel objects. Behavior in the groups of the EDA TSGs, the ultrasound assisted EDA TSGs and the EDA injection were close to the healthy group, showing an obvious tendency to explore the novel objects. Mice in the blank TSGs group showed the same performance as the model mice, exploring both the old and novel objects (Figure 8B).

The recognition index is in the range of 0–1, approaching 1 means inclined to explore the novel objects, while approaching 0 means inclined to explore the old objects. The recognition index of the healthy mice was close to 1, in line with the nature of mice liking to explore the novel objects. The recognition index of the model mice was lower than that of the healthy mice (*p* < 0.001), tending to explore the old objects. The recognition index in groups of the EDA TSGs (*p* < 0.01), the ultrasound assisted EDA TSGs (*p* < 0.001) and the EDA injection (*p* < 0.001) were higher than that of the model group, showing a trend to reach 1. Mice in the healthy group, the EDA TSGs group, the ultrasound assisted EDA TSGs group and the EDA injection group explored the novel objects more than the old objects, with an obvious trend but no significant differences. No obvious trend and statistical differences were found in the model group and the blank TSGs group (Figure 8C).

In general, the model mice had memory impairment and could not distinguish between old and novel objects. With the treatment of EDA TSGs, ultrasound assisted EDA TSGs and EDA injection, the mice spent less time exploring the old objects, and their memory recognition ability was clearly improved. Among them, ultrasound assisted EDA TSGs was superior to the use of EDA TSGs alone.

#### 3.6.3. The Elevated plus Maze Test to Detect Anxiety

The elevated plus maze can evaluate the anxiety state of the animal. All the healthy mice were moved freely in the open and the closed arm, almost arriving at the end of the open arm with no anxiety. The model mice moved more in the closed arm and rarely enter into the open arm, indicating anxiety behavior. Mice activities of the EDA TSGs group, the ultrasound assisted EDA TSGs group and the EDA injection group in open arms were increased. No improved effect was found in mice of the blank TSGs group (Figure 9A).

Compared with the healthy mice, the number and the time radio of mice entering the open arm in the model group (*p* < 0.01) and the blank TSGs group (*p* < 0.001) were decreased significantly. While above index in the EDA TSGs group, the ultrasound assisted EDA TSGs group and the EDA injection group were increased observably, which had significant differences compared with the model group (*p* < 0.001) (Figure 9B,C).

Overall, the treatment of EDA TSGs, ultrasound assisted EDA TSGs and EDA injection can attenuate anxiety behavior of RBI mice. However, there were no significant differences in the three groups, which could almost reach the level of the healthy group. 

#### 3.6.4. The Fear Conditioning Test to Detect the Reflex to Conditional Memory of Mice

The freezing behavior was used to describe the reflex to fear memory of mice. Rodents tend to maintain a defensive posture of immobility during fear, while animals with memory impairment exhibit actively due to loss of fear memory. The movement track of mice in the conditional fear box showed that the healthy mice concentrated in a certain position without movement, while the model mice had active spontaneous activities and their tracks were all over the box. Mice in the EDA TSGs group, the ultrasound assisted EDA TSGs group and the EDA injection group had decreased activities on the contrary to the blank TSGs group (Figure 10A).

The healthy mice showed more freezing behavior, which indicated that they had the fear memory and showed defensive posture spontaneously. Compared with the healthy group, the freezing time radio in the model group and the blank TSGs group were reduced significantly (*p* < 0.001), with no significant difference between the two groups, indicating their incomplete or disappeared fear memory. The freezing time ratio in groups of the EDA TSGs, the ultrasound assisted EDA TSGs and EDA injection was higher significantly than the model group (*p* < 0.001) (Figure 10B). The total distance of mice in the model group (*p* < 0.001) and the blank TSGs group (*p* < 0.01) were increased significantly compared with the healthy mice. The total distance in the groups of the EDA TSGs, the ultrasound assisted EDA TSGs and the EDA injection mice was lower than that of the model group (*p* < 0.001), but there was no statistical significance among the three groups (Figure 10C). 

The above data indicated that the treatment of EDA TSGs, ultrasound assisted EDA TSGs and EDA injection can improve the memory ability of the RBI mice, lead more freezing behavior. Moreover, ultrasound assisted EDA TSGs had a better therapeutic effect than the use of EDA TSGs alone and EDA injection according to the trends of statistical data. However, there was no significant difference, which may be due to the dispersion of data caused by individual differences.

#### 3.6.5. Histopathological Evaluation of Hippocampus in Brain

The hippocampus in the brain was mainly responsible for the formation and storage of memories, which mainly related to the dentate gyrus (DG) area, cornu ammonis (CA) areas including CA 1 and CA 3 in hippocampus [35]. In the healthy hippocampus, pyramidal neurons were arranged neatly and densely with a regular cell morphology and a large number of cell layers, and the synapses in CA 1 area were clear and orderly. Neuron cells of the model mice were arranged sparsely and irregularly, with increased nuclear pyknosis and hyperchromatism in the DG area and blurred and discontinuous synapses in CA 1 area. This indicated ionizing radiation could cause significant pathological changes and inflammation in the hippocampus. Compared with the model group, the pathological changes in the EDA TSGs mice, ultrasound assisted EDA TSGs mice and EDA injection mice were improved obviously. There was only a little phenomenon of nuclear pyknosis and hyperchromatism in the DG area of the EDA TSGs group. And no nuclear pyknosis and hyperchromatism were found in groups of the ultrasound assisted EDA TSGs and the EDA injection (Figure 11). This was consistent with the superior therapeutic effect of ultrasound on behavioral tests that evaluated cognition and memory, including the novel object recognition test and the fear conditioning test.

The treatment of the EDA TSGs, the ultrasound assisted EDA TSGs, and the EDA injection can alleviate the pathological changes of hippocampus caused by ionizing radiation, which further improved the decline of memory and cognition.

#### 3.6.6. Decreased Expression of MDA and IL-6 

MDA is a representative product of lipid peroxidation, which can indirectly reflect the severity of cell damage caused by free radicals [36]. Compared with the healthy group, the expression of MDA in model group and blank TSGs group were increased significantly (*p* < 0.01), showing more serious oxidative stress. The content of MDA in groups of the EDA TSGs (*p* < 0.05), the ultrasound assisted EDA TSGs (*p* < 0.01) and the EDA injection (*p* < 0.01) were significantly decreased compared with the model group. And the effect of ultrasound assisted EDA TSGs was better than that of the EDA TSGs (*p* < 0.05) (Figure 12A). Ultrasound can enhance the elimination effect of EDA TSGs on oxidative stress induced by RBI.

Inflammatory response is one of the important mechanisms of RBI [37]. IL-6 is a multipotent cytokine with extensive functions, which plays an important anti-inflammatory role [38]. Compared with the healthy mice, the expression of IL-6 in groups of the model and the blank TSGs was increased significantly (*p* < 0.01), which indicated have severe inflammatory reaction. In comparison, a significantly decreased (*p* < 0.01) of the level of IL-6 were showed in the EDA TSGs group, the ultrasound assisted EDA TSGs group and the EDA injection group. Furthermore, ultrasound assisted EDA TSGs was superior to the use of EDA TSGs (*p* < 0.05) alone and the EDA injection (Figure 12B). It showed that ultrasound could enhance the therapeutic effect of EDA TSGs by reducing inflammation. 

## 4. Conclusions

In this study, we prepared the injection preparation EDA TSGs and investigated the feasibility of ultrasound-assisted brain-targeted drug delivery. A series of evaluations showed that EDA TSGs had good therapeutic effects on RBI by relieving oxidative stress and inflammation, and the effects were enhanced with the addition of ultrasound. Notably, ultrasound improved learning and memory more than activity and emotions of animal in behavioral experiments. The prepared EDA TSGs overcome the problems caused by intravenous infusion, such as inconvenient administration route and long administration time in clinical practice. In conclusion, this study prepared a safe and effective formulation for RBI, preliminarily verified the efficiency of ultrasound-assisted drugs entering the brain, which provides new thoughts on the clinical treatment of RBI and more brain diseases.

## Figures and Tables

**Figure 1 pharmaceutics-14-02281-f001:**
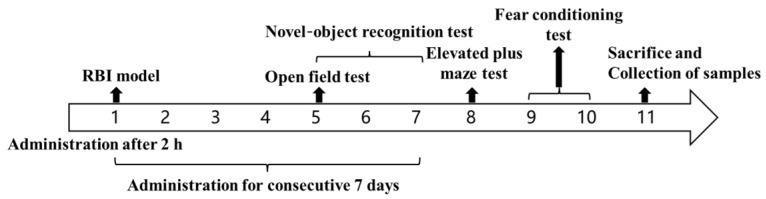
The schematic diagram of the pharmacodynamics test for the treatment of RBI.

**Figure 2 pharmaceutics-14-02281-f002:**
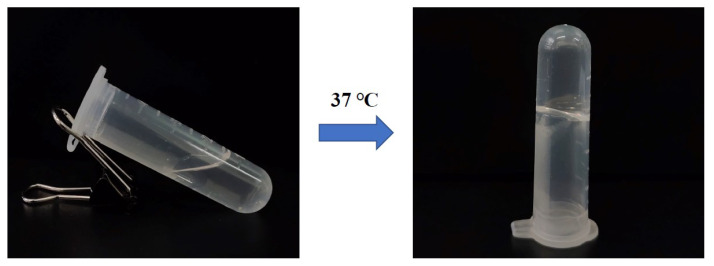
The morphology of EDA TSGs with the optimal prescription.

**Figure 3 pharmaceutics-14-02281-f003:**
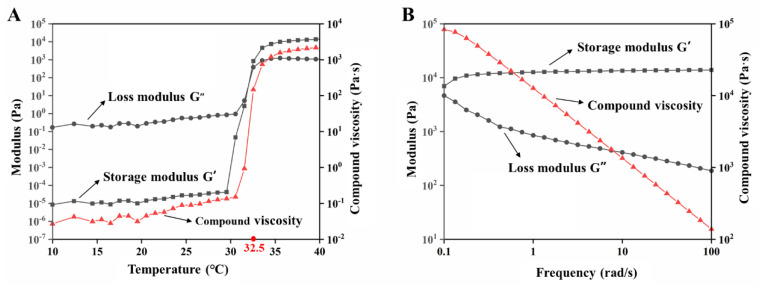
Rheological properties of EDA TSGs. (**A**) The profiles of storage modulus (G′), loss modulus (G″) and compound viscosity of EDA TSGs at 10–40 °C with the shear frequency of 6.28 rad/s, the strain amplitude of 0.05%, and the heating rate of 1 °C/min. (**B**) The profiles of storage modulus (G′), loss modulus (G″) and compound viscosity of EDA TSGs at body temperature of 37 °C with the range of 0.1–100 rad/s and the strain rate of 0.05%.

**Figure 4 pharmaceutics-14-02281-f004:**
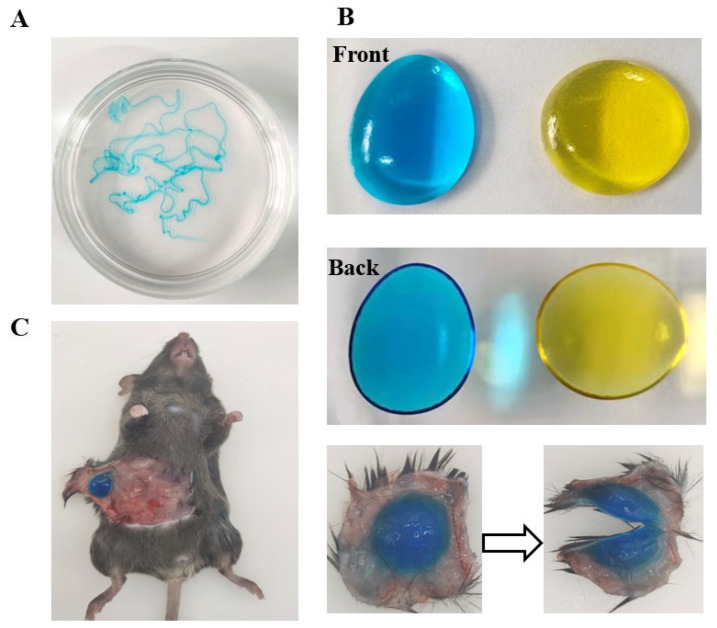
Good syringe ability and molding properties of EDA TSGs. EDA TSGs was injected into (**A**) water of 37 °C, (**B**) air of 37 °C and (**C**) mouse subcutaneous tissue.

**Figure 5 pharmaceutics-14-02281-f005:**
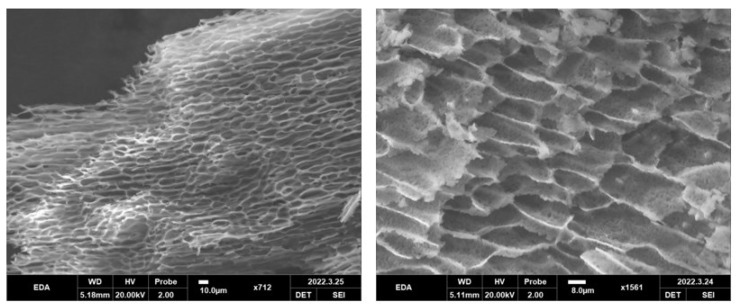
Micromorphology of EDA TSGs under SEM. An interwoven honeycomb structure with the pore diameter of about 10 μm was showed inside the gel.

**Figure 6 pharmaceutics-14-02281-f006:**
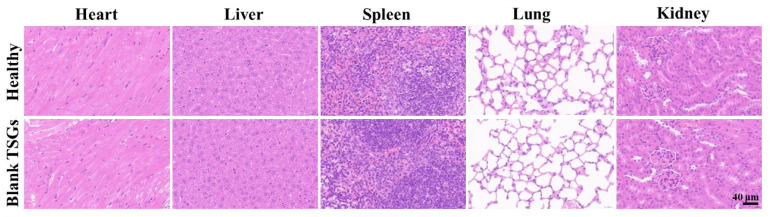
H.E staining of visceral organ in vivo. Scale bar = 40 μm. Healthy group: no operation; Blank TSGs group: continuous injection of blank TSGs for 7 d. No significant difference indicating safety in vivo of EDA TSGs.

**Figure 7 pharmaceutics-14-02281-f007:**
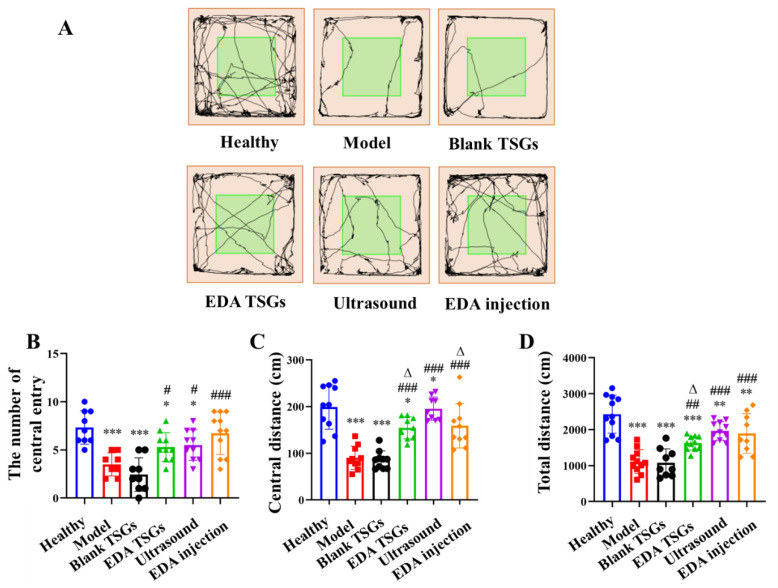
Comparison of spontaneous behavior, exploratory spirit and the mental tension of mice in each group by the open field test. The track map (**A**); Statistical data analysis of the number of central entry (**B**), central distance (**C**) and total distance (**D**). Data are represented as mean ± SD (*n* = 10). *: means significant difference, vs. the healthy group, * *p* < 0.05, ** *p* < 0.01, *** *p* < 0.001; ^#^: means significant difference, vs. the model group, ^#^ *p* < 0.05, ^##^ *p* < 0.01, ^###^ *p* < 0.001; ^Δ^: means significant difference, vs. the ultrasound group, ^Δ^
*p* < 0.05.

**Figure 8 pharmaceutics-14-02281-f008:**
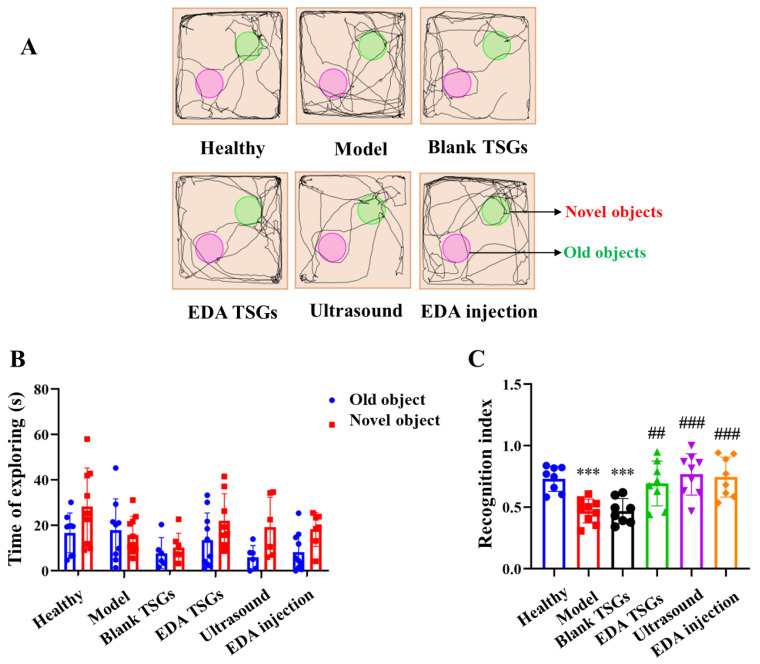
Comparison of learning and memory of mice in each group by the novel object recognition test. The track map (**A**); Statistical data analysis of the time of exploring (**B**) and recognition index (**C**). Data are represented as mean ± SD (*n* = 10). *: means significant difference, vs. the healthy group, *** *p* < 0.001; ^#^: means significant difference, vs. the model group, ^##^
*p* < 0.01, ^###^
*p* < 0.001.

**Figure 9 pharmaceutics-14-02281-f009:**
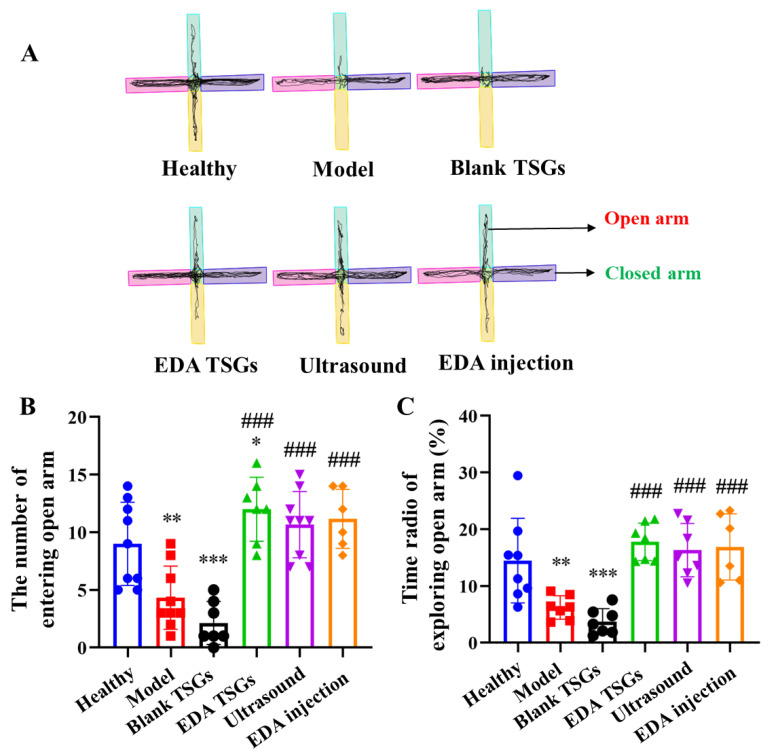
Comparison of anxiety behavior of mice in each group by the elevated plus maze test. The track map (**A**); Statistical data analysis of the number of entering open arm (**B**) and the time radio of exploring open arm (**C**). Data are represented as mean ± SD (*n* = 10). *: means significant difference, vs. the healthy group, * *p* < 0.05, ** *p* < 0.01, *** *p* < 0.001; ^#^: means significant difference, vs. the model group, ^###^
*p* < 0.001.

**Figure 10 pharmaceutics-14-02281-f010:**
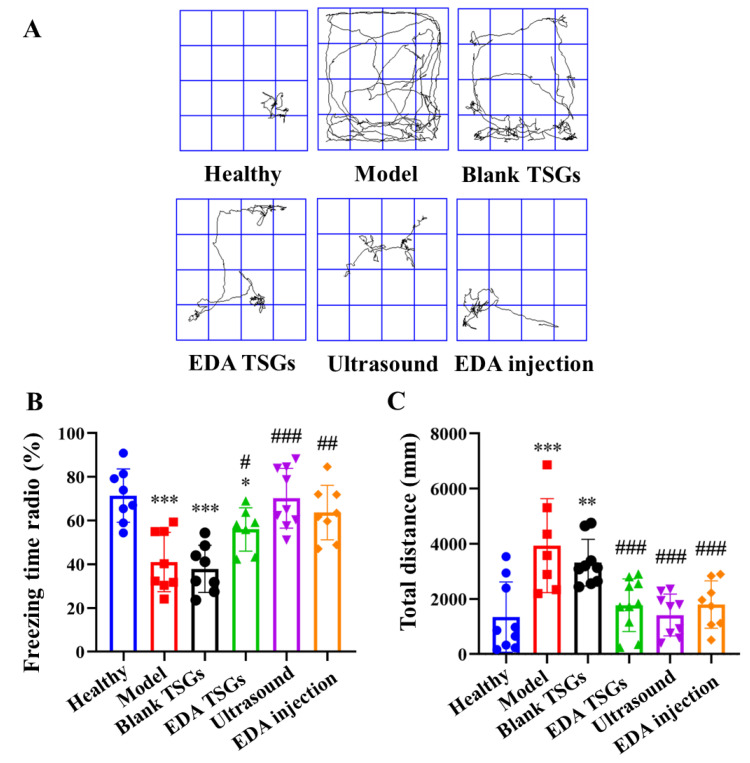
Comparison of reflex to conditional memory of mice in each group by the fear conditioning test. The track map (**A**); Statistical data analysis of the freezing time radio (**B**) and total distance (**C**). Data are represented as mean ± SD (*n* = 10). *: means significant difference, vs. the healthy group, * *p* < 0.05, ** *p* < 0.01, *** *p* < 0.001; ^#^: means significant difference, vs. the model group, ^#^
*p* < 0.05, ^##^
*p* < 0.01, ^###^
*p* < 0.001.

**Figure 11 pharmaceutics-14-02281-f011:**
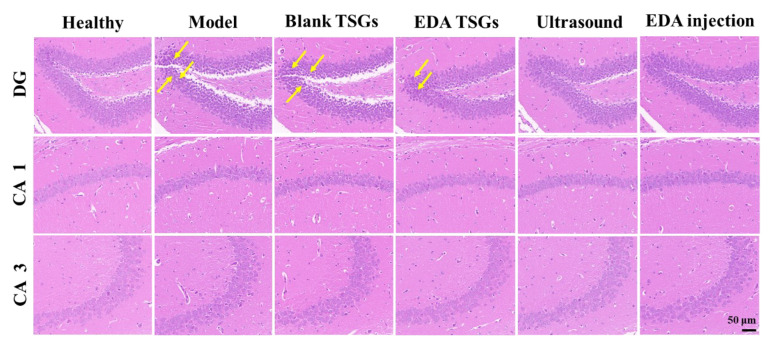
H.E staining of the hippocampus in each group. Scale bar = 50 μm. The yellow arrows represent the nuclear pyknosis and hyperchromatism parts of neurons.

**Figure 12 pharmaceutics-14-02281-f012:**
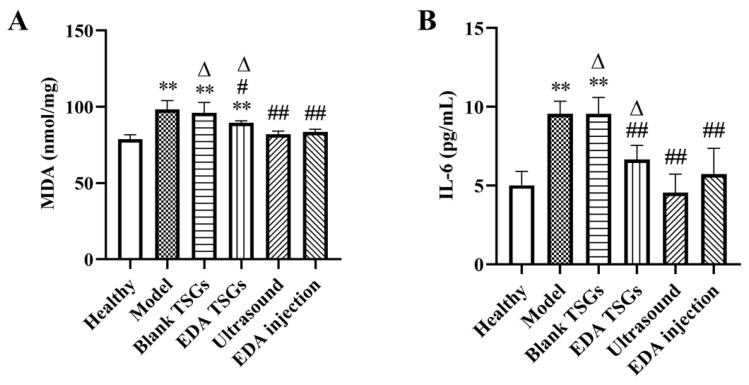
The decreased expression of MDA (**A**) and IL-6 (**B**) in hippocampus. Data are represented as mean ± SD (*n* = 3). *: means significant difference, vs. the healthy group, ** *p* < 0.01; ^#^: means significant difference, vs. the model group, ^#^
*p* < 0.05, ^##^
*p* < 0.01; ^Δ^: means significant difference, vs. the ultrasound group, ^Δ^
*p* < 0.05.

**Table 1 pharmaceutics-14-02281-t001:** Evaluation of gels with different prescriptions at different temperatures.

Number	Poloxamer (%)		Temperature (°C)					
	407	188	25	30	32	34	36	37
1	18	3	unset, good liquidity	unset, good liquidity	unset, good liquidity	unset, good liquidity	5 min unset	5.17 min solidify
2	20	3.3	good liquidity	5 min unset	3 min solidify	2 min solidify		1.33 min solidify
3	22	3.7	good liquidity	3 min solidify	1.67 min solidify	1 min 15 s solidify	50 s solidify	50 s solidify
4	24	4	poor liquidity, unset	1.67 min solidify	1 min solidify	50 s solidify	30 s solidify	30 s solidify
5	30	5	2 min solidify	50 s solidify	50 s solidify	40 s solidify	30 s solidify	20 s solidify
6	40	6.7	solidified at 16 °C					

## Data Availability

The data presented in this study are available on request from the corresponding author.

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
