# Peer review of "The Improved Brain-Targeted Drug Delivery of Edaravone Temperature-Sensitive Gels by Ultrasound for γ-ray Radiation-Induced Brain Injury"

_pharmaceutics, 2022, doi:10.3390/pharmaceutics14112281_

Round 1
Reviewer 1 Report
The authors here developed EDA-loaded temperature-sensitive gels for in situ application for radiation-induced brain injury. The study is very interesting, and the results are promising. However, the manuscript can be further improved to enhance its readability.
Introduction
’Radio-sensitivity refers to the body response to IR induced injury after low-dose IR exposure’
The definition of radiosensitivity is the susceptibility of body/cells/tissues response to IR. This does not limit to low-dose IR.
‘Brain is the most sensitive part of IR injury, which is prone to radiation-induced brain injury (RBI)’
This statement is not very specific as other stem cell-rich tissues such as skin and guts are also very sensitive to IR injury, and many will show earlier pathological effects than the brain due to their fast stem cell turnover rate.
‘This is important for protection of brain function to ensure combat effectiveness in long voyage and sea warfare.’
This conclusion comes from nowhere. Please clarify how RBI is relevant to long voyage and sea warfare.
Materials and methods
‘Ultrasound processing…the intensity was 0.6 W/cm2’
More information will be needed. Please specify which intensity it is (in terms of spatial and temporal considerations). Was the ultrasound applied in continuous mode or in pulse mode? What is the pressure? If the parameters were determined by previous studies, please cite the correct reference.
2.6. Establishment of RBI model and administration scheme
The description of experimental groups is very confusing in the current format. It would be best to list each group and each treatment condition specifically. For example, healthy group: no operation; model group: IR only…etc. For the ultrasonic assisted EDA TSG groups, were the 0.1 mL EDA TSGs injected immediately before or after ultrasound exposure? The model group should at least have been irradiated, right? What is the rationale for performing the behaviour tests at different time points while some are still under treatment? Where were the TSGs injected (near the neck? Limb?)?
2.9. The determination of MDA and IL-6 in the hippocampus: The homogenate was centrifuged at 4 °C with 12000 r/min for 10 min,
Please describe the spin speed in g (RCF) or provide the centrifuge model. RPM is a relative value and will result in different g-forces on different centrifuges.
2.10. Data analysis: Statistical differences between groups were analyzed with one-way ANOVA.
Please indicate which post hoc analysis was used to determine the significance.
3.1. EDA TSGs can achieve solution-gel transformation
The second paragraph contains repetitive information as in the previous section and should be removed or moved to the materials and methods section. It would be nice to show the gelation time of different prescriptions here, not just the successful one.
3.4. Microstructure of EDA TSGs
Does the pore size/microstructure change after EDA loading?
3.5. In vivo safety of EDA TSGs: Continuous injection of blank TSGs after 7 d.
Was the experiment using EDA-loaded or blank gel? The subheading does not fit to the context.
Page 17: In general, the healthy mice had memory impairment and could not distinguish between the old and the novel objects.
Should it be the model mice?
It would be nice re-arrange Figures 7A, 8A, 9A, 10A, 11, and 12 to follow the same order as the column group in Figure 7B.
Figure 11 did not show any difference across groups. Images with higher magnification may be needed to reflect the claim in the context.
It will be nice to have a summary of what each behaviour test was studied (learning, anxiety, memory, etc) in the opening of 3.6 Pharmacodynamics evaluation for the general readers. Also, further links to the behaviour test results with the histology result should be included in section 3.6.5.
Reviewer 2 Report
In this study, the authors prepared the injection of EDA and thermoresponsive gels in combination with ultrasound therapy to provide relief to RBI. The authors used various behavioral tests and biochemical analyses in support of their method.
The topic should be interesting to the readership in the field of pharmaceutics. Unfortunately, the manuscript is found minimalistic e.g. lacking proper figure legends and method details, and sometimes confusing due to the combination of techniques used. With the main concern that the methods claimed do not provide any substantial benefit over already existing FDA-approved injections, I am reluctant to support its publication unless additional data and writing revisions are provided.
Major concerns are given below in detail -
1) Many sentences in the introduction are missing relevant references. Please provide relevant references wherever applicable.
2)Methods section is severely lacking the exact details of amounts, brand names, and procedures that are necessary for the proper reproduction of the data.
3)Results section should generally discuss the actual findings and their interpretation. However, many times, authors have given protocols that should be placed in the methods section (e.g., page 10, last paragraph).
4) Results
a. 3.1 - How was the optimal prescription selected? Data supporting the choice of this particular prescription over others should be provided.
b. Figure 3A – provide error bars
c. 3.2.1 – provide data to show that the semi-solid-state gel can continuously release the data
d. Figure 4C – how many mice were tested for this? Is this the representative image or the only image showing in vivo data? Provide specifics.
e. Figure 6 – It would have been nice to include immunostainings for inflammatory cells.
f. Figure 6 –the data are given only for healthy and model animals. Please provide additional data for the rest of the conditions as well.
g. Figure 7 - one-way ANOVA can be misleading when performed on 2 individual groups each time often accentuating the significance. A better way to do this is to compare all the groups simultaneously and then report the significance values.
h. For all histopathological studies, please provide enlarged (zoomed-in) views. These zoomed-out images are highly inefficient in judging differences if there are any.
5) All the figure legends are highly minimalistic and quite insufficient in providing information. Please revise.
6) Naked EDA injection should be the actual control as TSGs would provide an additional advantage over naked EDA injection. It looks to me that EDA injection is sufficient to provide beneficial effects in all cases. TSG + EDA does not offer a significant benefit over naked EDA making the entire study near inconsequential. If the authors can show that EDA+TSGs are indeed better than EDA only, then only the study may hold some importance for publication.
Round 2
Reviewer 2 Report
I feel that my suggestions have been taken into account satisfactorily and I now support its publication in this modified form.